

# The influence of soil on the impacts of burst water mains on infrastructure and society: A mixed methods investigation

Timothy S. Farewell[1], Simon Jude[1], Oliver Pritchard[2]

[1] School of Water Energy and Environment, Cranfield University, MK43 0AL, UK
[2] Arup, Blythe Gate Blythe Valley Park, Solihull, B90 8AE, UK

*Correspondence to*: Timothy S. Farewell (t.s.farewell@cranfield.ac.uk)

**Abstract.** Society relies on infrastructure, but colocation and interdependencies make infrastructure systems vulnerable to cascading failures. This study investigated cross-infrastructure and societal impacts of burst water mains, with the hypotheses that 1) burst main-triggered cross-infrastructure failures are more common in sandy soils and 2) mixed-methods approaches

are more beneficial than pure data analysis for understanding the wide-ranging impacts of these events. When water mains in sandy soils burst, pressurised water can create sub-surface voids and abrasive slurries, contributing to further infrastructure failures. To investigate the role of soil in hosting cascading infrastructure failures, maps of soil sand content for England and Wales were created. Analysis of the infrastructure impacts arising from burst mains combined; 1) spatio-temporal clustering and analysis of infrastructure failure data, 2) meta-analysis of web-based media reports of burst mains impacting on other

networks, and 3) workshop discussions and structured interviews with infrastructure industry experts. The workshop, interviews and media reports produced a greater depth of information on the infrastructure and societal impacts of cascading failures than the spatio-temporal data analysis. Cross infrastructure impacts were most common on roads, built structures and gas pipes, and occurred at a higher rate in soils with very high sand contents.

## 1 Introduction

The socio-economic and physical wellbeing of society is increasingly dependant on infrastructure services (Lloyds Register Foundation, 2015; Guikema, 2015; Defra, 2013) and infrastructure assets (e.g. pipes, cables, roads, substations, pumping stations and buildings) are commonly co-located. Consequently, a failure of one asset (e.g. a burst water main) may lead to failures in proximal networks (e.g. damage to, and flooding of gas networks). Complex infrastructure failures can be cascading, escalating or have a common cause (Rinaldi et al. 2001). They can occur at a range of spatio-temporal scales and affect both

physical and socio-political infrastructure.

Multi-infrastructure failures often result from a single failure in the crowded and heterogeneous array of co-located, aged and modern, interconnected and semi-automated infrastructure systems (Pritchard et al. 2014a). These systems operate with physical, geographic, functional, cyber, policy, and informational dependencies and interdependencies (Rinaldi et al., 2001;



Zimmerman, 2004; Dudenhoeffer et al., 2006). These close relationships can make infrastructure vulnerable to complex failures. Potential for the initial infrastructure failure is influenced by both inherent infrastructure factors (age, material, diameter, joint technology, workmanship, co-location, pressure management, investment) and environmental factors (soil, vegetation, extreme or rapidly changing weather). Rapid or extreme environmental changes often expose the vulnerabilities of

aging and deteriorating infrastructure networks.

The heterogeneity of infrastructure assets, networks and the soil environment produces further complexity for infrastructure operators and regulators tasked with providing robust and resilient levels of service (Rinaldi et al., 2001; Rogers et al. 2012). With the limited awareness amongst infrastructure asset managers of system-of-systems thinking, which is rarely employed in

asset risk assessments, and the "siloed" approach of governments and regulators, understanding of infrastructure interdependencies is often lacking (Young and Hall, 2015; Defra, 2011; Jude et al., 2017; Street et al., 2017; Committee on Climate Change, 2017).

An example of how water bursts can impact on other infrastructure networks occurred in Matlock, Derbyshire, UK, (BBC,

2013a). Here, flooding from a burst main closed two roads, disrupting transport across the city. Escaping water formed a void under the road surface, into which a water-company van fell, fracturing a gas main. The gas leak forced the evacuation of 25 homes, water and sediment flooded the gas network, and the County Hall suffered flood damage (including to official records) and was closed for days. This one burst damaged roads, gas networks, and buildings. It impacted government functions and required police and fire service resources. Whilst direct costs of this complex failure totalled many tens of thousands of pounds,

indirect costs were much higher.

Natural hazards have different frequencies, impacts and spatio-temporal scales. A considerable body of literature exists surrounding acute environmental hazards such as flooding (e.g. Bowering et al, 2014). However, less research explores more complex, and often chronic, forms of soil related natural hazards and related infrastructure failures (Defra, 2011). Such hazards

pose significant risks to infrastructure systems that may be currently underestimated by stakeholders. Because risk-perception is often linked to past experience (Taylor et al. 2014), the impact of low-frequency events with moderate-high impacts may be underestimated by infrastructure operators, as they are not high on organisations risk registers.

Soils support infrastructure, yet some are prone to forms of ground movement including clay shrink-swell, sand washout, and

peat shrinkage (Pritchard et al., 2014a; 2015a, 2015b). While the process of soil movement is relatively well understood, little is currently known about the likelihood of complex infrastructure failures resulting from water pipe bursts in different soil types.



Sandy soils have greater than 70% sand-sized particles (0.06-2mm) and cover less than 20% of England and Wales (Cranfield University, 2016). They are susceptible to water-assisted erosional processes (Brink *et al*. 1982). Water escaping from buried pipes can form voids, removing the structural support normally offered by soil to infrastructure (bridging), while also forming abrasive slurries which are highly damaging to plastic pipes (Majid et al., 2007).

The second UK Climate Change Risk Assessment identified cascading infrastructure failures as the highest risk facing UK infrastructure. The assessment recommends greater consideration of subsidence risks to infrastructure, and improved risk-information sharing between infrastructure operators (Dawson et al., 2016).

10 This paper presents an interdisciplinary scoping study exploring the impact of sandy soils on the impacts of burst water mains on physical infrastructure (electricity, natural gas, water, wastewater, transport and telecoms), public service infrastructure, (government, emergency services, healthcare and education), and wider socio-economic functions. The hypotheses are twofold: 1) That sandy soils are more likely to give rise to multi-infrastructure failures due to their non-cohesive structure and composition of large, abrasive particles that, under the release of high pressure water, can lead to damage of proximal

15 infrastructure. 2) That a mixed methods approach will be more beneficial than purely quantitative data analysis for understanding the wide ranging impacts of these events. Three methods and multiple sources of evidence are used. Discussion focuses on the impacts of these failures on different infrastructure, and wider society, as well methodological approaches and the risk management implications of this research.

**2 Methods**

20 The control of soil on the impacts of burst water mains on proximal infrastructure was investigated with three methods, in three study areas. These include: 1) quantitative spatial data analysis of industry-reported infrastructure failures for a local authority region (1a) and water company region (1b) in the East of England; 2) meta-analysis of media-reported multi-infrastructure failures across England and Wales; and 3) a workshop and interviews with UK infrastructure practitioners. The different regional study areas (Fig. 1) were guided by the availability of infrastructure data.



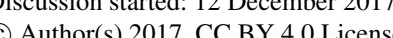

**Figure 1 – Maximum sand content map at 80 cm depth for England and Wales, with study areas for the different methods, and media-reported cascade failures, which are described in detail in Appendix A.**

## 2.1 Sand content maps for England and Wales

Maps of soil sand content were produced for England and Wales using the 1:250,000 scale National Soil Map and Land Information System (LandIS) data (Cranfield University, 2016). Soil texture varies with depth, and across the soil mapping units, so multiple sand content maps for minimum, maximum and weighted average sand content at 0, 40, 80 and 100 cm depth

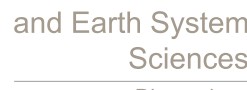
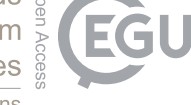


were created to reflect the variability of soil sand content associated with common infrastructure burial depths. In this paper, for consistency, the maximum sand content at 80 cm depth (which approximates water pipe depth) is used in all analyses.

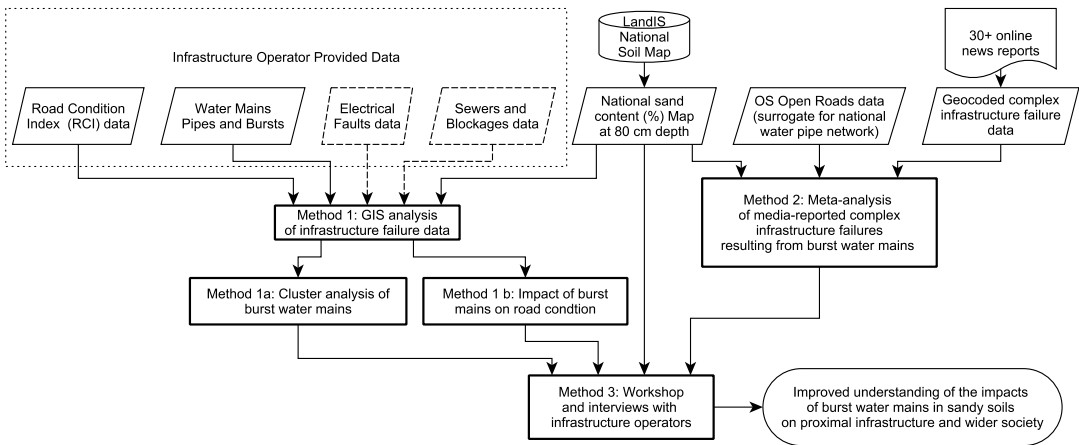

Figure 2 - The flow of data and information between the three methods. Note: Dashed lines indicates datasets which were collected but not of suitable quality for use in the analyses.

## 2.2 Method 1: GIS spatio-temporal cluster analysis of infrastructure failures

An analysis of infrastructure failures was undertaken in two study areas in the East of England for which detailed asset and failure data was available (Fig. 1). Infrastructure asset and failure data was obtained from water, sewerage, road and electrical infrastructure operators. Electrical and sewerage fault data were discarded as they were of insufficient accuracy for spatial analyses. Thus, the analysis focussed on assessing the impact of burst mains on roads (Method 1a) and the impact on other water pipes (Method 1b).

### 2.2.1 Method 1a: Measuring the impact of burst water mains on roads

The impact of burst water mains on co-located roads was investigated across the county of Lincolnshire (Fig. 1) by analysing road quality survey data collected before, and after, reported bursts. Each burst was buffered by 50 m to identify the road segments under the '*potential influence*' of the burst main (0-50 m, grey circles, Fig. 3), and an area which was presumed '*beyond influence*' of the burst (50-100 m, blue circles) but representative of similar soil and road materials.

Annual (2008-2013) Road Condition Index (RCI) SCANNER data was provided by highways engineers at the County Council. RCI is measured on a scale from 0 (good) to 315 (failed) (Wallis, 2009; UK Roads Board, 2011; Pritchard et al, 2014b, 2015b). Roads with RCI >100 require maintenance. RCI change after a burst was calculated and analysed in the light of soil sand content at 80 cm depth. Both degraded road conditions (positive RCI) or improved conditions (negative RCI) could indicate an impact from a burst main.




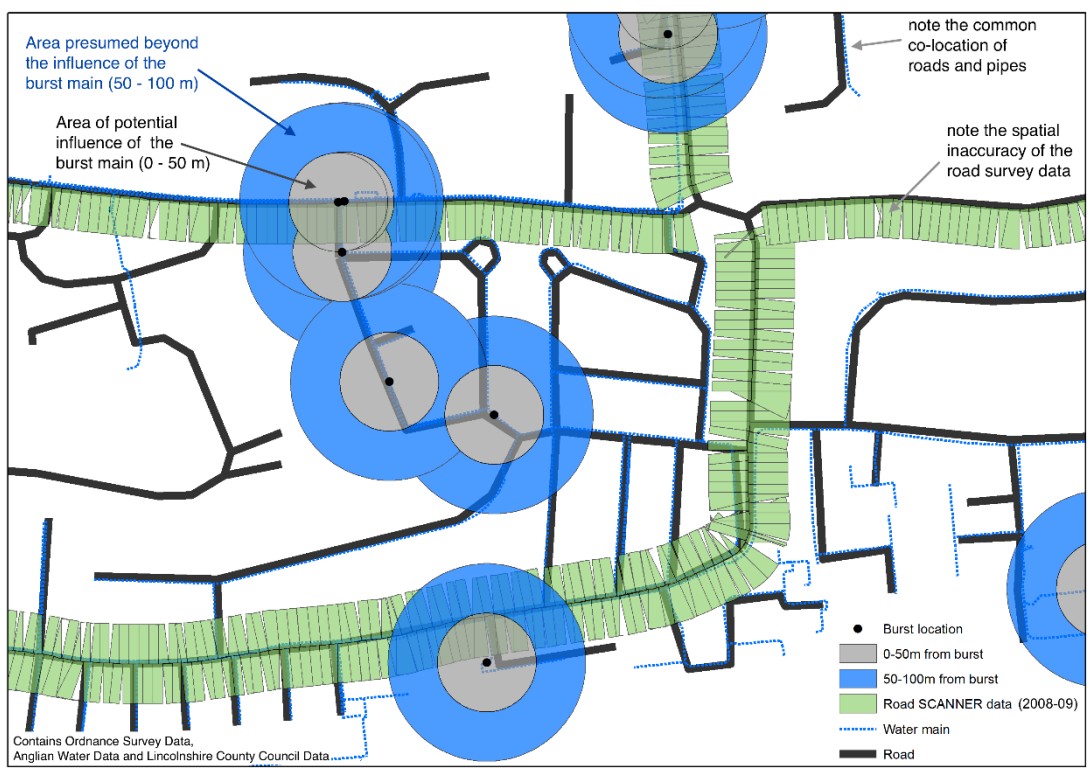

**Figure 3 – Example region showing road sections which potentially have been influenced by a burst, and the similar road sections which are presumed beyond the area of influence (Method 1a).**

5 RCI data was not available for all roads, in all years, and the opposite sides of the road were typically surveyed on alternate years (Fig. 3). The road condition survey area polygons are 10 m wide, and extend beyond the road footprint. To mitigate this spatial inaccuracy, a count of these polygons was used in this context simply to calculate a change in condition.

The rate of failure was calculated by dividing the number of bursts in clusters by the total number of bursts in each sand decile.
10 By their nature, larger spatio-temporal windows have higher rates of clusters. Therefore, for comparison, the rates have been normalised by dividing the rate, by the sum of all the rates, for each panel in the graph (Fig. 7). The calculation used is:

$$\text{Normalised rate} = (\text{clusters}_s / \text{bursts}_s) / (\Sigma_t (\text{clusters}_t / \text{bursts}_t) \qquad [1]$$

Where:

clusters $_s$ = the number of clustered bursts within a sand decile

bursts $_s$ = the total number of bursts within a sand decile

clusters $_t$ = the total number of clustered bursts in this spatio-temporal window





bursts $_t$ = the total number of burst in this spatio-temporal window (the whole dataset)

### 2.2.2 Method 1b: Measuring the impact of burst water mains on other water mains

To test if burst mains in sandier soils were more likely to trigger subsequent bursts, clusters of water main bursts were identified using expanding spatio-temporal windows: ((distance: 2, 5, 10, 30, 100 m) (time: 1, 5, 10, 100, 365 days)). These windows were chosen to capture the different failure patterns. For example, the smaller windows (e.g. 2 days, 5 m) may identify multiple bursts triggered directly by the bursts; through force transmission down the pipe, sand abrasion, or failures triggered by a common cause. Longer temporal windows may identify impacts stemming from secondary impacts. For instance, a road surface weakened from cutting to access the pipe, or due to voids, may increase variations in traffic-loading forces on pipes, and so, increase the risk of failure. The number of burst clusters were compared with maximum soil sand content at 80 cm. The rate of failure of all bursts per km pipe, by sand content was also calculated. 50,901 bursts from Anglian Water between 2004 and 2015 were used in the analysis.

### 2.3 Method 2: Meta-analysis of media-reported burst water main events

A meta-analysis of over 30 UK local media reports between 2009-2017 was employed to identify the complex forms of failure arising from burst water main events. This time period was chosen for the widespread availability of UK web-based articles from this time. Google searches including key words such as "water main", "burst", "road", "electricity", "phone, "gas", and "sewer" provided articles. The date and impacts of the burst mains were recorded (summaries are provided in Appendix A). Bursts location was estimated from the location descriptions in the articles were geocoded with www.gridreferencefinder.com. The geocoded data was imported into ArcGIS and attributed with information from the sand maps.

To calculate normalised rates of failure (e.g. bursts per 100 km of pipe), the length of water mains in each soil map unit across the UK (Fig. 1) was estimated using the OS Open Roads data ("A", "B" and "Unclassified" Roads) as a surrogate (OS, 2016). This results in a 7% underestimate of the length of pipe in East Anglia (39669 km roads vs 43000 km pipes). This error is sufficiently small for the purposes of this research, and no spatial bias in the linear infrastructure data was observed. Spatial bias may (or may not) occur in the locations of the events, using this approach. For example, if a particular newspaper has identified cascading failures in the past, it is more likely that they may report these issues again. Conversely, if such failures happen weekly, these events may be under-reported as they are no longer "newsworthy". Future research should consider accuracy assessments of these approaches in more detail. In this scoping study, the assumption of no spatial bias has been made.



### 2.4 Method 3: Cross-infrastructure workshop and interviews

A stakeholder workshop, involving representatives spanning water, electricity distribution, gas distribution and highways sectors was used to elucidate the key impacts of burst water mains on other infrastructure. The workshop employed a trained facilitator and used a semi-structured experience-sharing discussion format. Discussions focussed on experiences of sand

washout impacts on infrastructure assets, service provision and risk management challenges.  Follow-up semi-structured one-to-one interviews with workshop participants further explored particular issues of interest. Interviews were also held with local authority, rail and telecom representatives who were unable to attend the workshop. The workshop notes and interview transcripts were analysed using thematic analysis to identify key themes.

### 3 Results and Discussion

This section provides combined results and discussions from all three methods. First, the control of soil sand content on the rate of cascading infrastructure failures is discussed. Then, the impact of burst water mains on other infrastructure, and wider society, is investigated. Implications of this scoping study for risk management are then discussed. Finally, the mixed-methods approach is assessed. The media meta-analysis (Method 2) and the workshop / interviews (Method 3) revealed insights into the wider impacts of water mains on other infrastructure that were hidden from the pure spatial data analysis (Method 1).

The workshop and interviews provided a framework for extracting infrastructure operators' perspectives on cross-infrastructure impacts of burst mains. Network operators described these failures as low frequency, but moderately high impact events. The importance, and difficulty, of cross-infrastructure communication and co-working was identified (Dawson et al., 2016) but the value of cross-sector regional task groups was asserted.  Details from these discussions are illustratively

incorporated in this section. For brevity, citations of comments from the workshops and interviews, and the meta-analysis media articles, are omitted from the text. The media articles are summarised in Appendix A.

Where figures include error bars, they show the 95% confidence intervals for the poisson mean. This interval is calculated by transforming a symmetric 95% confidence interval (CI) for the logarithm of the mean.

### 3.1 The control of soil on the location of cascading infrastructure failures

The rate of water mains bursts does not appear to be controlled by soil sand content. The four bands of sand content in Fig. 4 all have a very similar rate of bursts of between 0.97 and 1.05 bursts per km of pipe. However, the meta-analysis indicated that sand content does play a controlling role in the likelihood of an initial burst going on to impact on other infrastructure.  A higher rate of media-reported cascading infrastructure failures was observed in sandy soils (Fig. 4).



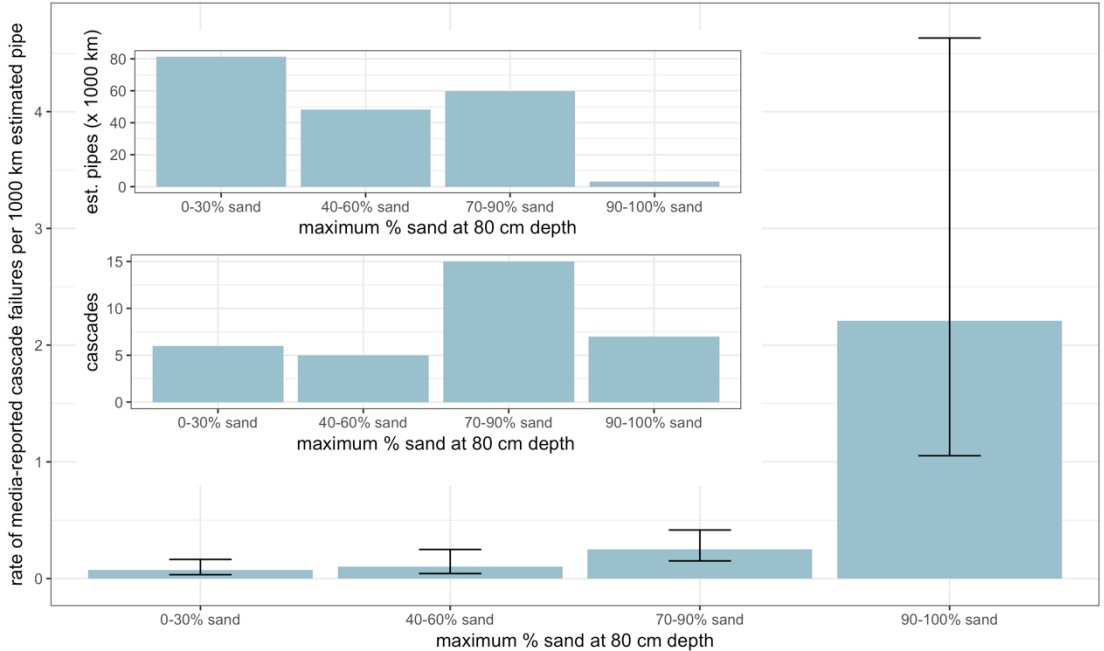

**Figure 4 - Rate of media-reported cascade failures, normalised by pipe distribution (estimated from national A, B and Unclassified road length) in each soil type. Error bars (95% Confidence Interval for the poisson mean).**

## 3.2 The impacts of burst water mains on other infrastructure

The meta-analysis of media reports identified 33 multi-infrastructure events across England and Wales between 2009 and 2017 (Fig. 1). The articles (summarised in Appendix A) provided detailed information on the impacts of burst mains on infrastructure and wider society (e.g. school and hospital closures, length of traffic delays, amount of bottled water delivered, and the emotions of those impacted by the events). The impacts of burst water mains on infrastructure and wider society are summarised in Fig. 5 and Table 1. Co-located roads and gas pipes were the most commonly affected infrastructure.

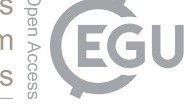

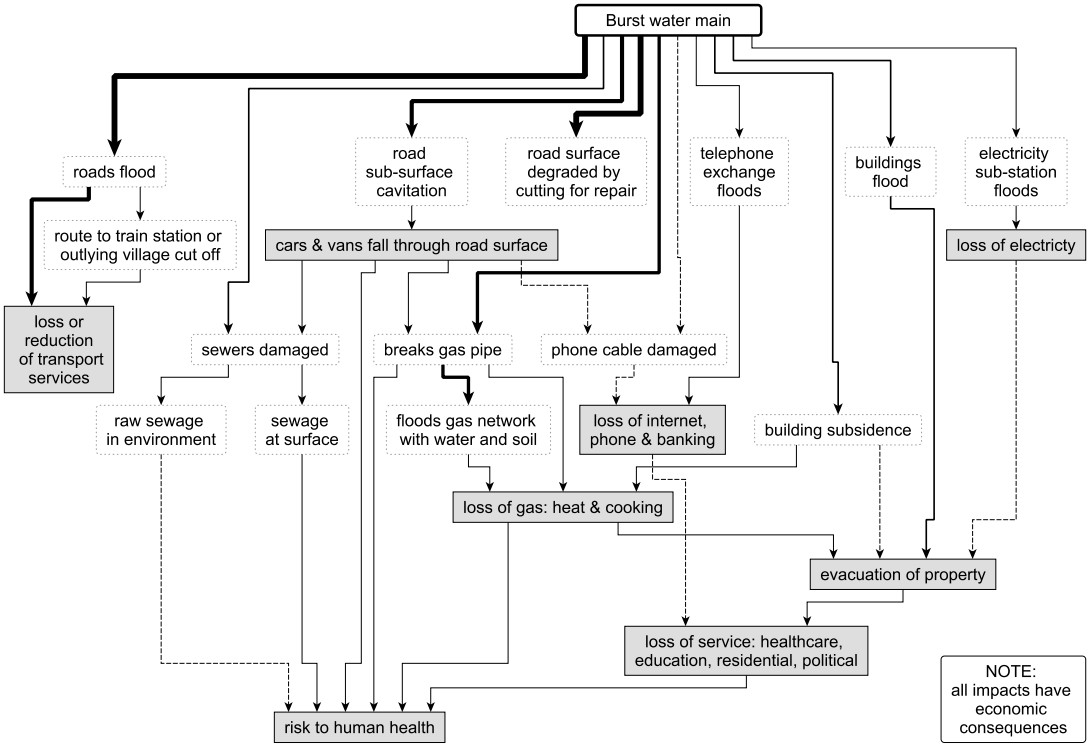

**Figure 5 – Summary of impacts from burst water mains on other infrastructure and wider society. Schematic diagram based on analysis of 33 media reports, workshop discussions and interviews, showing impacts to other infrastructure and society. Line width represents the relative frequency of the impact.**



| Infrastructure | Reports | Impacts |
|---|---|---|
| Road | 21 | Flooding, surface damage, sinkholes (+/- vehicles in them), traffic delays, closure |
| Houses | 10 | Loss of water, loss of gas, flooding, sewage flooding, evacuation, subsidence, extensive cracking |
| Gas | 8-11 | Loss of gas, fractured pipe, flooded and sediment in gas mains |
| Buildings | 6 | Flooding of county hall, schools closed, hospital wards and accident and emergency (A+E) department closed and patients transferred. Shops closed. Lamp posts unstable. |
| Sewers | 3 | Blocked sewers leading to foul flooding. Pumping station filled with sand. Tankers required to pump sewage. Sewer collapse. Raw sewage in garden. |
| Health | 3 | Health suffering due to cold exposure, sewage in gardens, A+E closed, patients moved. Toilets out of action. |
| Electric | 2-5 | Loss of electronic payments. Facilities unable to open. |
| Water | 2-3 | Loss of water, second pipe repair in close proximity. |
| Telecoms | 2 | Loss of phone and internet services (including no credit card payments at a supermarket for many hours.) |

**Table 1 - Summary of impacts on other infrastructure from burst water mains (from analysis of media reports).**

Note: where reports indicate a range (e.g. 8-11), this is due to uncertainty in the descriptions provided by the media.

5 **3.2.1 Roads**

Flooding and damage to roads are common direct impacts from co-located pipes. Void formation under the road surface can also impact on safety (e.g. vehicles falling through the road surface into voids). Minor and local roads are more likely to be impacted by water mains failures than major roads, as minor roads are more commonly underlain by water pipes, and are less well engineered. However, examples where major roads have been impacted include a burst-formed void on a major road in

10 Kent costing a water company £640,000 in remediation, and causing a 25 day road closure. Burst mains have also flooded motorways causing significant disruptions.

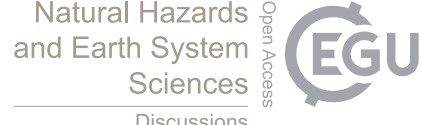



Road damage or flooding can extend travel times and distances and can result in reputational damage to the water and highway operators. Diversions in rural areas of up to 48 km were identified. Major voids will lead to longer road closures, and greater socio-economic impacts.

5     In the water company region (Fig. 1), 93% of minor (B) roads have pipes within 16 m of the centre line of the road. It is logical that a failure in the pipe network will impact directly on the road, through void formation, or indirectly through road-cutting to access and repair the pipe. To test this, the change in road condition was assessed (prior to, and after a burst) using annual road condition (RCI) surveys for 232897 road segments *"potentially influenced"* (0-50 m) by a burst main and 262140 segments deemed *"beyond influence"* (5-100 m) of the burst (Fig. 3).

With only one exception, for all roads, the mean RCI change was approximately 0 (Fig. 6), with consistent interquartile range (IQR) for all roads. However, roads built on sandy (70-90%) soils, within 50 m of a burst main, showed greater spread in the change in road condition, which indicates that greater remedial work is required to roads following a burst in sandy soils.

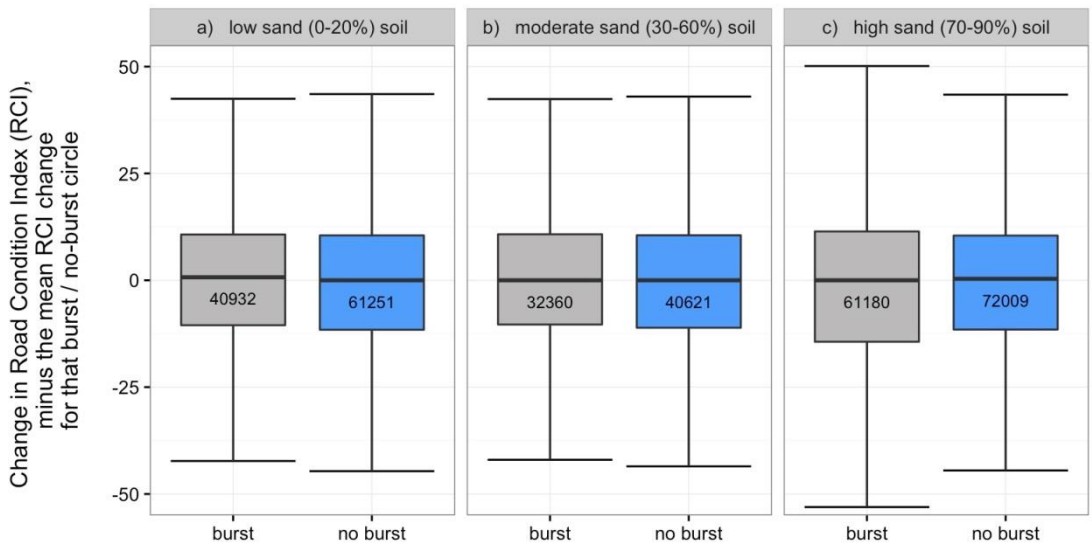

**Figure 6 - Comparison of the change in RCI before and after a burst water main (minus the mean RCI change for the circular sample area (Fig. 3). The numbers on the box plots represent the number of analysed road segments. Whiskers: Range excluding outliers (IQR +/- 1.5*IQR).**

20     Thus, bursts in sandy soils appear more likely to change the road surface condition than bursts in other soil types (Fig. 6). Even if the road is not damaged by the burst and water pressure, pipe repairs commonly require cutting the road surface to access the failed pipe. Highways authorities within England and Wales report that such cutting and trenching impacts the structural integrity of the road, and potentially reduces the roads service life by 30% (Asphalt Industry Alliance, 2016) and this was also





reported independently at the workshop. Cuts represent physical lines of weaknesses in a previously solid, load bearing surface. Because the road surface is repaired to a lower standard than that of the original road surface, the shorter serviceable life of the road leads to higher maintenance costs for councils. Trenching will also provide preferential hydrological pathways for water compared to the surrounding ground. Local highways engineers noted that road cuts may be contributory factors to

subsequent water pipe failures at the same location which are commonly reported within 1 year.

Where non-catastrophic cavitation occurs over an extended period of time (due to a small water leak from mains, or frequent infiltration / exfiltration of sewers), a commonly reported symptom is road profile change, which can provide an early warning of issues beneath the road. Multiple media reports described the initial misdiagnosis and repair of small road surface

deformations only to find a much larger hole the next day.

### 3.2.2 Ports and railways

Ports and railway stations represent critical access nodes for international and national transport. The vulnerability of the access routes to the Ports of Felixstowe and Lowestoft were discussed in the workshop, as parts of these key transport routes are on sandy soil. If roads are closed due to cavitation from a burst main (or tidal surge, as occurred outside the Lowestoft train station

in 2014) then access to the ports / railway would be severely restricted. The economic and transport consequences of port closures are severe.  As well as preventing access to these transport nodes, burst mains can also affect railway infrastructure itself. In August 2016, a burst water main contributed to the collapse of a railway embankment and bridge in Leicestershire disrupting rail journeys for thousands of passengers for a number of days.

### 3.2.3 Gas distribution pipes

Gas pipes can be damaged by water mains as a result of: (1) the pressure of the water itself, (2) water + soil mixed to an abrasive "sandblasting" slurry, or indirectly through (3) cavitation and subsequent damage by vehicles or road surface collapse. Such failures commonly cause neighbourhoods to lose gas supply (300, 400, 800 homes in media reports).

The cost of pipe-repairs is insignificant compared to the cost of removing water and sediment from the pipes. In some reported

cases, up to 10,000 litres of water and debris needed to be pumped from the gas network. Removing water and sediment is a complex process leaving properties without gas for extended periods of time. In one burst-triggered gas network failure, supplies to 250 customers were lost for 7 days due to the valve-less low pressure gas networks. These pipes required repeated digging (damaging the road) to (1) insert a camera to find the blockages, (2) to isolate the main, and then (3) to physically isolate each property. There are additional regulator-costs associated with loss of service, and potential health risks for

vulnerable people due to lack of heating.





Health risks are higher when gas in buildings (GIBs) events occur, following a leak. These can occur through migration of gas through the soil into houses, but also when water enters gas mains through cracks. As more water enters the pipe, the gas pressure will drop for short periods to a point where some pilot lights can extinguish, leaving gas entering into unlit boilers. These types of failures are hard to predict. Gas meters and boiler valves can also be damaged by water and debris in the

network.

### 3.2.4 Buildings and houses

Public and private buildings are commonly impacted by water mains failures, both directly (e.g. flooding) and indirectly. In one burst near Bristol, 8000 homes lost water supply for 3 days. Properties can also lose gas supply. In one example, 25 homes were evacuated due to a large gas leak. When sewers are blocked due to sediment ingress, sewage can enter houses through

the toilets. Property subsidence has also been reported following a burst main near a house on sandy soil as a result of cavitation. This led to cracks opening up in the walls in the winter, and health impacts for the vulnerable residents were reported.

### 3.2.5 Other water mains

While multiple water mains failures were only reported 3 times in the media analysis, the GIS cluster analysis identified that 2-3% of bursts were co-located with another burst within 5 metres and 5 days of the original burst. For clusters within 2 metres

and 1 day, a slightly higher rate of failure was observed for pipes in the sandiest soils (Fig. 7). A water company reported higher rates of multiple pipe failure due to sand abrasion for softer polyethylene pipes than metallic pipes.

Using 50,901 burst water main records, the spatio-temporal windows identified clusters for between 1% (1 day, 2 m radius) and 45% (365, 100 m radius) of the bursts. When the patterns in those two windows are compared, (Fig. 7) the smallest

windows shows a higher rate of clusters in the sandiest soils, but also a high rate for the low-sand soils – probably indicative of common cause failures associated with clay soils (e.g. corrosivity or shrink-swell). These clusters in the low-sand content soils increase with expanding windows reflecting the larger number of bursts in these corrosive and movable soils.



**Figure 7 - normalised rate of (burst cluster) / (all bursts within expanding spatio-temporal window), by maximum soil sand content at 80 cm depth. Error bars: 95% CI for the poisson mean. Higher bars indicate more clusters of bursts per initial trigger burst. For clarity, only 9 of the 25 spatio-temporal windows are shown, but the full graph is provided in Appendix B.**

5   **3.2.6 Sewers**

Sewer impacts from burst mains include blockages, and flooding by sewage of roads and gardens. Such incidents are unpleasant and carry associated health risks. When properties are without sewerage, tankers are required.

As sewers do not require the same structural integrity as gas and water mains, and have joints every few metres, they are

10   vulnerable to exfiltration of sewage, and infiltration of water and particles. The change between high and low external pressures



can lead to void formation around the sewer. Increased water pressures can come from burst mains or natural events such as storm surges, or high rainfall events. Sandy soils are more likely to be washed into the sewers than clays and loams. A large water company invested a large amount of money cleaning sand from the sewers in Lowestoft, only for the sewers to fill up with sand again following the next storm surge.

A water company that manages both water distribution and sewerage networks reported that voids around sewers are more problematic than around mains pipes. When reported, voids can be filled with a resin. If left unchecked, the structural integrity and flow pathways of the sewer can suffer as the sewers settle into the void. This in turn can increase the probability of a subsequent blockage, which can in turn lead to sewer flooding.

10 ### 3.2.7 Electrical distribution

Flooding from burst mains is a potential risk to urban electricity infrastructure, where substations and electrical equipment are commonly located in basements or underground recesses. One below-ground substation had suffered two floods in two years resulting in £1m costs and subsequent relocation of equipment. Any disruption to electricity supply has wide impacts, including on small IT networks.

Only minor impacts on electricity distribution networks from sand-washout events were identified, with 12% of media reports mentioning electricity distribution impacts (Fig. 5). An electrical Distribution Network Operator attributed this low impact rate to buried electricity cables having sufficient flexibility to accommodate a loss of ground support, and that the higher voltage cables were buried at greater depth. However, older forms of lead-paper insulated cables exhibit limited flexibility 20 and are thus more vulnerable. Another reason for the resilience of the electricity networks is that they are reconfigurable, with supplies rarely interrupted for more than a few seconds, anywhere other than single source nodes of the network.

Electric cables are most commonly damaged by "third party strikes" when water companies and gas companies dig down to repair or replace their assets. Notable advances have been made by utilities to avoid these strikes and the associated risk to 25 human life, and additional damage, but they still occur as the electric cables often sit on top of water mains.

### 3.2.8 Telecom cables

Telephone cables appear resilient to burst main impacts, possibly due to the prevalence of overhead lines in older residential areas. Only one example of a burst main resulting in telephone disruption was categorically identified by the media analysis. However, in this instance, when the phone lines were cut off, a very large supermarket was prevented from accepting credit 30 card payments until the lines were repaired.




### 3.3 The impacts of burst water mains on wider society

The socio-economic implications of burst mains range from simple repairs of the infrastructure, to more complex impacts such as increased travel times, loss of work, and disruption to businesses through loss of footfall or disruptions to electronic payments. If roads serving isolated communities are closed, the impact of even a week of lost earnings can be catastrophic for

small businesses. Schools and hospitals cannot open without water, and numerous examples of such closures were identified.

Whilst health is rarely affected directly by burst mains, secondary impacts were identified. Examples include closure of hospital units and the movement of vulnerable patients to other hospitals, raw sewage in gardens and subsidence leading to the formation of cracks in houses with associated heat loss and implications for the health of older residents. When gas mains are

ruptured, houses may be evacuated to minimise health impacts. When cars become trapped in holes in the road there is potential for significant injury or death. While it is the duty of infrastructure operators to minimise risk, there are also longer term socio-economic and liability costs if human health is affected. Furthermore, any major disruption to infrastructure service provision can result in public relations and customer satisfaction impacts.

### 3.4 Implications of this research for risk management

Sand washout is not the most common, or damaging, soil related geohazard (Pritchard et al 2014a). Due to the distribution of sandy soils (Fig. 1), regional trends can be observed. This research identified mixed levels of awareness of sand washout risk from infrastructure operators. While some local authorities (particularly those in sandy soil areas) have dedicated teams to address this issue, most utilities only deal with these events on a case-by-case, reactive, basis. Although some of the impacts

of these events have been considerable, it was noted that the low-frequency of such events make them difficult to consider as part of many asset management plans.

Infrastructure monitoring incurs large costs and is often unfeasible across an entire network, so reactive responses to infrastructure failures are common. Nevertheless, it was noted that the use of soil maps and other geohazard datasets to identify

assets and communities at risk from washout and other events would be a first step most infrastructure organisations could take to identify (and then potentially mitigate) their exposure to these risks. Such maps can inform decision-making, help prioritise areas for increased levels of maintenance, or faster response times, and inform asset management plans.

The infrastructure failure data analysed did not provide the severity or scale of the impact. One burst main may cost a nominal

amount to repair, but one which impacts on other infrastructure systems can have significant costs associated. Both bursts are currently represented by one record each in the company bursts database. Utilities may wish to record the severity and scale / cost of the impact in their relational spatial databases to identify areas of their network which commonly are more expensive





to fix. The importance of collecting and maintaining highly accurate spatial data for assets and failures is asserted, if later data-analysis is to be undertaken and meaningful results provided to inform future decision making.

Information sharing around infrastructure interdependencies between utilities is often only undertaken on a 'need to know' basis. Because of a focused remit on their own infrastructure, low levels of information sharing on environmental hazards and risks occurs even between similar networks in the same geographic region. However, many countries are seeing a transition towards large parent companies owning multiple utility companies (e.g. in the UK Cheung Kong Infrastructure Holdings Limited largely owns Northern Gas Networks, Northumbrian Water and UK Power Networks, and also owns a strategic stake in the Southern Water Group.) As a result, where appropriate, information sharing between these companies operating in the same area is encouraged by the parent company. Independent operators working in similar regions may take part in local infrastructure groups, or national infrastructure resilience networks.

Many utilities stated that their awareness of systemic vulnerabilities, risks and interdependencies was less than ideal, and expressed their desire to better understand the societal risks beyond that of their own network concerns and liabilities. A desire for greater quantification of the impacts of these type of low frequency events on levels of service and resilience was expressed. While this scoping research begins to address these desires, there is potential for a more thorough analysis of these types of failures, using and building on the approaches used in this research.

### 3.5 Assessments of the mixed-methods approach

This scoping study sought to quantify the impact of burst mains on other infrastructure, and determine if a mixed methods approach provided more information than a pure GIS data analysis of reported infrastructure faults and failures. Indeed, the value of the meta-analysis of media reports and expert knowledge distilled from workshops and interviews quickly became apparent.

The spatial data analysis quantified the impact of burst mains on proximal water mains (Method 1a), and on the condition of the roads (Method 1b) and how these vary by soil type. The unavailability and/or inaccuracy of many infrastructure datasets did not permit the desired identification of many cascading infrastructure failures. Industry-reported data did not describe wider societal impacts, nor the scale or cost of the failures. Significantly, the industry GIS data was usually restricted to the location, date, and repair type undertaken. In contrast, the meta-analysis of media reports (summarised in Appendix A) provided qualitative descriptions of both infrastructure failures and the impacts on health, economy and people (Method 2). Because media reports tend to focus on the larger bursts than little leaks, the results can be expected to be more dramatic than usual. However, analysis of these reports identified that the rate of these more dramatic failures is higher in areas of sandy soils. Because of the depth of information gleaned from this approach, media meta-analysis is encouraged for other studies of low frequency, moderate impact local environmental risks, but further work on the removal of any spatial bias from such





reporting is recommended. Social media feeds may also serve as a crowd sourced dataset for identifying these types of failure. The workshop and one-to-one interviews with infrastructure owners and operators (Method 3), captured detailed perspectives on these cascading infrastructure failures and their impact on service delivery, costs, responses and management plans. The combination of these three methods led to more quantifiable, descriptive and useful results than would have been possible if

each method was used in isolation.

Adopting a mixed methods approach to research does bring its own challenges. The default skillset of a data scientist may not be sufficient to cover the more qualitative aspects of research, so a multi-disciplinary team is required. For example, vastly different approaches to assessing the accuracy or bias of data are required for GIS data, compared to that obtained from local

media reports, or in a workshop discussion.  In a mixed methods investigation, it is recommended that a shared vision of a successful outcome of the research is defined near the start of the investigation so that all methods contribute towards this goal.

## 4 Conclusions

Diverse examples of the cross-infrastructure impacts from burst water mains have been identified and discussed. Cascading infrastructure failures, while occurring in many soil types, appear to be more than three times as common in soils with high

sand content (Figure 4). While the investigations undertaken in this research have focussed on the UK, the same principles will apply in any country where sandy soils are present (e.g. Majid et al., 2007). The types of failures described tend to be low frequency, moderate impact events. Due to asset co-location, roads and gas pipes are the infrastructures most commonly affected by burst water mains. There are substantial direct and indirect economic costs of these events.

The impact of burst water mains on other infrastructure can be long-lasting (reduction in the structural integrity of a road) or costly to repair (e.g. removing water and sediment from a flooded gas network). Burst mains can also impact on the wider society; disrupting healthcare, increasing travel times, and closing local businesses, government operations and schools. The costs of these societal impacts are rarely quantified, and are typically borne by affected individuals. Wider discussions around cascading failures are of relevance to regional infrastructure and resilience groups.

Critically, the research illustrates the potential value of mixed methods approaches to investigate such complex infrastructure hazards and risks.  Whilst the geospatial data analysis of infrastructure failures provided insufficient information to fully address questions about the impact of burst mains on proximal infrastructure, the meta-analysis of local news stories provided rich information relating on the cascading impacts of burst water mains. Furthermore, the direct input from infrastructure

operators through the workshop and interviews obtained valuable information on their views on these risks to their infrastructure resilience. Thus, the authors believe that the mixed methods approach holds great potential for infrastructure research, but such mixed approaches do require careful development and evaluation.



Marker (1998) argued that earth science is generally underused in spatial planning. Soil maps, similar to those developed in this research can help infrastructure companies identify assets in soils vulnerable to sand washout, and other more common soil-related geohazards (Pritchard et al 2014a). Clear identification of the hazards present in a local area will enable informed

decision making. Vulnerable assets can be identified, assessed and repaired or proactively replaced to minimise cascading impacts.

**Author contribution.** Farewell and Jude conceived the study. All authors were involved in data collection including the workshop, with Farewell and Jude analysing the results. All authors contributed to the manuscript.

**Competing interests.** The authors declare that they have no conflict of interest.

**Acknowledgements.** This work was supported by NERC [NE/M008320/1] and EPSRC/ESRC [EP/I01344X/1; EP/K012347/1]. The sponsors played no role in the study design or the collection, analysis or interpretation of the data, nor

any role in the writing of the report or decision to publish these results. Enquiries regarding the data generated in this study should be sent to researchdata@cranfield.ac.uk. This research would not have produced the insights without the generous provision of data, time and expertise from the utilities (water, gas, electrical distribution) highways (local and national) telecoms and railway organisations who provided invaluable input. The authors would also like to thank Caroline Keay and Ann Holden for their assistance in the creation of the new sand washout potential map and the cluster analysis tool, and Daniel

Farewell and Vern Farewell for statistical advice.

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



**Appendix A**

| ID | Date | Sand % | Summary of Media Report | URL |
|----|------|--------|--------------------------|-----|
| 1 | 23/02 /2009 | 93 | 15 cm PVC main burst. Damaging road surface. Police involved. 69 houses off water. Fire engine called to pump water. Bottle water. Took 20 hours to fix pipe. | http://www.northnorfolk news.co.uk/news/cromer _water_main_fixed_1_5 33520 |
| 2 | 02/11 /2009 | 80 | Large diameter main (76 cm) burst. 8000 homes without water. 18 schools closed. Bristol Water and Red Cross handing out water. 19 people rescued by dinghy, and spent the night in a church hall. Huge hole in road. Gardens destroyed. | http://www.bbc.co.uk/ne ws/uk-england-bristol- 29373980 |
| 3 | 06/10 /2011 | 95 | Burst main floods gas pipe. 650 houses off gas. 80,000 L of water removed from gas network. Significant damage to gas meters and appliances. Gas company supplied electric hobs and heaters to affected homes. Set up a customer centre at the local church. | http://www.bbc.co.uk/ne ws/uk-england-dorset- 29362291 |
| 4 | 27/09 /2012 | 95 | Gas network flooded with water. 400 homes affected, some for more than 2 days. Engineers required to carry out safety checks, and reconnect gas. Customers off gas for 24+ hours are financially compensated. | http://www.bbc.co.uk/ne ws/uk-england-dorset- 29929187 |
| 5 | 10/10 /2012 | 87 | Burst water main. Major incident declared at Scunthorpe Hospital. No drinking water & toilet flushing affected. Patients told not to attend A+E if possible. | http://www.itv.com/news /calendar/update/2014- 10-05/water-supplies- restored-to-hospital-in- lincolnshire |
| 6 | 20/12 /2012 | 47 | A main road and footpath in Lincoln are closed for two days after a burst water main. | http://news.bbc.co.uk/1/h i/england/lincolnshire/83 37851.stm |




| 7 | 06/04 /2013 | 23 | Burst main - car fell through road. Closed road leading to gridlock. 100 m of road to be reinstated. 30 cm main. Road closed for 3-4 days. Water supplies off. | http://www.getsurrey.co.uk/news/local-news/burst-water-main-leaves-gaping-4813168 |
| 8 | 26/04 /2013 | 11 | Burst main leads to void under road. Car becomes stuck in hole. Both lanes closed. Many roads in Walton gridlocked. Police called to scene. Council made aware. | http://www.getsurrey.co.uk/news/surrey-news/car-trapped-sinkhole-opens-walton-7936966 |
| 9 | 01/11 /2013 | 92 | 1.2 m x 1.2 m void under road. Not sure if it is caused by gas leak, or if the void caused the gas leak. Road closed for a number of days. Smelling gas for a month before the hole was discovered. | http://www.getsurrey.co.uk/news/surrey-news/road-closure-after-sinkhole-appears-7259207 |
| 10 | 06/01 /2014 | 87 | Car stuck in hole on A320. No disruption to water supply. Water company paying car insurance claim. Resurfacing road. Road closed for 1 days. Police closed road. 15 inch main. | http://www.getsurrey.co.uk/news/surrey-news/car-becomes-lodged-burst-water-6983196 |
| 11 | 16/01 /2014 | 87 | Ground collapsed under car, driver missed work. Lamp post unstable, electric supply isolated. Water supplies to area affected. Schools in Chertsey and Woking closed. Centre for disabled kids and adults closed. Extensive damage to roundabout, 3-4 day closure. 38 cm main. | http://www.getsurrey.co.uk/news/local-news/video-ground-just-collapsed-under-4809243 |
| 12 | 21/03 /2014 | 95 | Burst main. Closed road to facilitate repairs. Also damaged electricity duct. Found second water leak. Interim repair first, and full repair in a few weeks. | http://www.worcesternews.co.uk/news/10120224.Road_is_reopened_after_leak_in_pipes/ |





| 13 | 09/04 /2014 | 95 | 100-year-old large burst main- 1000's people off water. Significant road damage (A6). Road closed for more than a week. Busy commuter route near M1. | http://www.bbc.co.uk/news/uk-england-leicestershire-25619109 |
| 14 | 13/04 /2014 | 64 | Old mine tunnel collapse which also damaged sewer pipes. The main impact here is the economic impact on local businesses. One road closure has lost a butcher 20% of his business, and a fish and chip shop has had no passing trade. The road will take more than a week to repair. | http://www.bbc.co.uk/news/uk-england-cornwall-25975404 |
| 15 | 17/04 /2014 | 95 | Another car in A320 hole. 38 cm Victorian main. Local traffic congestion. PR issues now because of repeated problems with cars falling through roads. | http://www.getsurrey.co.uk/news/surrey-news/affinity-water-continue-patchwork-repairs-7000547 |
| 16 | 22/08 /2014 | 39 | Burst water main. Void formation - driveway collapse and household subsidence. Cold air coming through cracks, with claimed health impacts. Insurance loss adjustors and legal representatives will be agreeing the next steps. | http://www.worcesternews.co.uk/news/10354380.We_re_scared_our_houses_are_collapsing/ |
| 17 | 26/09 /2014 | 23 | Burst main closes main road in Swindon for many days. 30 m of road surface damaged by main burst (30 cm). | http://www.bbc.co.uk/news/uk-england-wiltshire-22312127 |
| 18 | 28/09 /2014 | 95 | Burst main (caused by BT contractors). Floods 90 homes, 9 flats and retirement homes. Fire crews involved - bottled water supplied. 30 gas company staff involved. Gas supplies cut to 86 homes, removed 10,000 L of water from the gas network. Lots of sand in pipes too - complex engineering process to vacuum out the pipes. Giving out heating equipment. | http://www.getsurrey.co.uk/news/local-news/river-running-down-road-burst-6263464 |



| 19 | 30/09 /2014 | 0 | Mains bursts. Floods roads. Blocks sewers. Debris washing towards main road roundabout. Gas leak. Fire crew and local council workers both involved to unblocked sewers. Police closed road. 100s homes off water. | http://www.mynewtown. co.uk/viewerheadline/Art icleId/8437 |
|---|---|---|---|---|
| 20 | 04/10 /2014 | 87 | Burst main breaks gas pipes and flooded gas network. 755 properties with no heating or hot water for days. 100,000 L of water removed so far. 150 properties off gas for extended period. Distributing fan heater and warming plates. Working with local authority social services. Washing facilities for people provided by sports centres. | http://www.walesonline. co.uk/news/local-news/gas-disruption-nantyglo-leaves-hundreds-7660937 |
| 21 | 14/10 /2014 | 88 | Burst pipe. Floods 5 homes. Cut electricity supply and telephone lines. Bad PR for Yorkshire Water. | http://www.thetelegrapha ndargus.co.uk/news/local /airelocal/11757812.Resi dents_face_flood_after_p ipe_gives_out/ |
| 22 | 04/11 /2014 | 11 | 5 x 3.5 X 1 m deep sinkhole in garden from burst main. Destroyed pavement and garden. Began as a small hole in kerb. County council called, but no one came so police called. Police put up barriers. The next day, huge hole full of water. Anglian Water fixed the pipe when called. | http://www.northantstele graph.co.uk/news/top-stories/sinkhole-opens-up-at-bottom-of-cottingham-garden-1-6509161 |
| 23 | 18/11 /2014 | 87 | Taxi stuck in 1.5 m wide pothole caused by burst water main in Hampstead. Road affected for a number of days. | http://www.hamhigh.co.u k/news/environment/taxi _stuck_in_pothole_cause d_by_burst_water_main_ in_hampstead_1_394661 4 |
| 24 | 28/11 /2014 | 93 | Road closed for 3 days after burst cause road to collapse. Tree has fallen into hole. 10 houses off water for 6 hours, but took much longer to fix the pipe, as the actual leak was > 1 km away from the damaged road. Diversions in place. | http://www.kentonline.co .uk/sevenoaks/news/road -collapse-leads-to-closure-30803/ |





| 25 | 09/01 /2015 | 0 | 60 cm hole in road. Caused by burst main / or "drainage pipe". Old mines also present in the area. | http://www.stokesentinel.co.uk/2ft-sinkhole-Fenton-road-caused-burst-water-pipe/story-21070606-detail/story.html |
| 26 | 26/01 /2015 | 76 | 1.8 x 2.7 m wide, 1.8 m deep void. Destroyed road. Gardens flooded with sewage. Cascading failure damages proximal water mains (more bursts) and sewers (damage). Sewage pumping stations no longer working as sand and gravel in the pumps. Exposes gas pipes - . Tankers pumping sewers "day and night". 35 properties affected. | http://www.kentonline.co.uk/romney-marsh/news/huge-sinkhole-opens-near-homes-27226/ |
| 27 | 27/01 /2015 | 72 | Burst main fixed rapidly, but road remains closed to allow tarmac to set. Buses running 60 minutes late. | http://www.bournemouthecho.co.uk/news/11633506.Burst_water_main_repaired_but_traffic_misery_continues_for_motorists_in_Branksome/?ref=mr |
| 28 | 29/01 /2015 | 40 | Burst main closes road. Water coming out of BT manhole. Water flowed onto carriageway & freezes. Traffic backed up 3 km. Gridlock on surrounding roads. 1 primary school closed. | http://www.sussexexpress.co.uk/news/county-news/a272-closed-at-buxted-1-6557919 |
| 29 | 03/02 /2015 | 67 | Burst main floods allotments. Complex fix as gas pipes and power cables close to water main. 15 cm main. Some properties off water. Bottled water provided. | http://www.ilfordrecorder.co.uk/news/environment/burst_pipe_in_woodford_green_leaves_residents_without_water_and_an_allotment_flooded_1_3930119 |





| | | | | |
|---|---|---|---|---|
| 30 | 04/02 /2015 | 89 | Burst main floods gas network. 297 houses off gas. 200 homes off water. Heaters and portable cookers provided. | http://www.examiner.co. uk/news/west-yorkshire-news/hundreds-homes-moldgreen-dalton-tandem-7855557 |
| 31 | 06/02 /2015 | 86 | Burst main - sandy torrent of water, flooded 3 homes, turned road into "sodden beach". Water up to knee height - water up to 1 m high in houses. No water. No power. Road blocked for repair by police, fire crews required to pump water. 15 cm main. | http://www.edp24.co.uk/ news/environment/photo _gallery_burst_water_pi pe_floods_road_in_dersi ngham_1_3851694 |
| 32 | 08/05 /2015 | 40 | Burst main forms a void under road into which a Severn Trent van falls, cracking a gas pipe leading to the evacuation of 25 homes. Tens of thousands of pounds of flood damage. Roads closed for many days. Local council records flooded and offices closed for many days. | http://www.bbc.co.uk/ne ws/uk-england-derbyshire-22050687 |
| 33 | 19/11 /2015 | 100 | 38 cm main burst. Traders, charities and community centres closed, especially those with toilets, and cafes. Delays to repair of water supply because of a large electronic sign in a concrete plinth with a power cable rising through the middle, requiring specialist teams. Requested residents not to use dishwasher or washing machines to preserve water in tanks. | http://www.getsurrey.co. uk/news/surrey-news/woking-loses-water-supply-due-6941235 |
| 34 | 31/01 /2014 | 39 | Burst main. Flooding driveways and gardens. Traffic delays | http://www.getsurrey.co. uk/news/surrey-news/gardens-drives-flooded-after-a320-6860355 |