# Peer review of "How the impacts of burst water mains are influenced by soil sand content"

_Natural Hazards and Earth System Sciences, 2017_

## Referee Comment (RC1) · T. Acland (Referee) · 2 Jan 2018

Very good and some surprising findings that lead to valuable conclusions. Although only a comparatively small number of water mains bursts are involved with other utilities, these have some significant impacts. Ofwat said they would be monitoring the level of impact (e.g. in the news) as a regulatory (ODI, Outcome Delivery Incentive) performance/asset management check on water companies. It will be very nice if the water, gas, electric and telecoms companies had and shared their criticality information. Water mains have no national rules for criticality as there are for sewers so this needs to be addressed. There has to be a cost saving advantage as well. Typically

only 20% of a water company pipeline scheme cost is the physical infrastructure, the rest being construction costs which may be shared. But first the industries have to realise the common risk and consequence they have. Even saving some of the high consequence and repetitive failures must be beneficial to everyone.

1. Page 5 Sect 2 Line 10. The sewer data is explained as being of insufficient accuracy. It is also likely that these repairs were detected and resolved much later in time from the initial event. There are lengths of sewer still in operation although in poor condition, and stay that way for a long time.

2. Page 7 Sect 2 line 20. Total mains length will include 'sundries' e.g. hydrant and wash out 'legs', short pipe section associated with valve complexes. The 7% difference in length may be greatly reduced by discounting mains less than something between 1 - 5m in length. You will probably end up spot on!

3. Page 12 Sect 3 Line 20 and Page 13 Line 6. Are there proportionately more clamp burst repairs in the sand clusters than cut repairs? Cut repairs are usually more serious repairs and are more likely to be responded to quicker as an emergency. Clamp repair bursts may have been running for a much longer time and cause more sub-surface damage.

4. Page 14 Sect 3.2.5. This is interesting. The development of wide tolerance repair fittings and the Regulatory/Customer service need for rapid supply restoration may be leading to water companies to repair rather than replace short sections of mains. We have definitions for sewer criticality but no national guidelines for water main criticality. Further there is no nationally agreed sharing of these details between the utility companies.

5. Page 17 Sect 3.3 Line 6. Whilst most hospitals have second/alternative supplies another significant issue is the closure of schools. The impact on students and families will be significant.
6. Page 11 Sect 3 Line 5. May be say that minor roads have a more historical or appropriate level of engineering instead of saying they are less well engineered.

---

## Referee Comment (RC2) · K. Mertens (Referee) · 18 Mar 2018

This manuscript investigates the consequences of burst water mains for society by means of a mixed methods approach. The topic of bursts in water mains is an interesting issue of cascading events and their consequences for society. The manuscript is clearly written. I think the manuscript would benefit from some reframing of the hypotheses and improvements in the presentation of the data and results. Also, if possible, some further analysis and discussion of the results would be appreciated to make sure the manuscript really adds to the existing literature. After revisions I think the manuscript is suitable for publication in NHESS. In the following sections I first present

my main comments, followed by some minor issues.

Main comments General framing: The manuscript is currently presented as a scoping study which has two, very distinct hypotheses which require very different methodologies. On the one hand it is said that the authors want to investigate the relation between soil texture (sand) and the frequency of cross-infrastructure failure. On the other hand they want to test and evaluate a mixed methods approach for this kind of studies. I think these two objectives are too different to fit into one research paper, and certainly so if this research paper does only present the result of a scoping study. The current analysis does not test the second hypothesis. I therefore suggest dropping this second hypothesis and just mention that the purpose is to investigate the influence of soil on the consequences for society by combining different methods. Doing so the manuscript illustrates the value of mixed methods approaches, but does not test a hypothesis as to whether a mixed method is better or worse than another approach. It still allows you to discuss the merits of such an approach in the discussion and conclusion of your research paper.

Alternatively, the objective could be to investigate the value of a mixed method approach to study this kind of things. The technical analysis of the correlation with soil can then be presented as one of the different methods or as a case study.

It would be good if the objectives and research questions were made explicit. Now, I did only find the hypotheses and had to deduce the objective of the manuscript from it myself.

Introduction

Page 3, line 15: How is this hypothesis tested? By presenting this as a hypothesis it is suggested that the value of this approach will be tested. That is currently not being done.

Methods

Page 3, line 19: I would appreciate a section on the data, or more information on the data at least. How many bursts do you have for your study region (on the map we currently only see those that have led to a cascading event)? How many of these are in sandy soils? What percentage of those that are not on sandy soil have led to cascading events? This comment also holds for the news items. You mention 30+ reports, but is this for 30 different events? How many of these were on sandy soil?... After the section 'methods' I remain with the question whether you have information about all bursts or only about those that led to a cascading event. Do you not have a problem of selection-bias in your sample (eg soil types that have a better water evacuation not leading to cascading events being underreported in the number of bursts)?

Page 4, line 5: I am a little surprised about this sand map. First, at several places in the manuscript it is mentioned that a soil map was created (abstract, or page 20, line 2), while I guess soil maps were already existing. For this manuscript a reclassification of soil groups was just ran based on sand content. I might be wrong, but this does not seem very new to me and it shouldn't be presented as a main contribution of the manuscript. The manuscript makes other contributions. If more than that was done, this should be mentioned here.

Page 5, line 2: It is mentioned that for consistency sand content at 80 cm depth was used. But were other depths also tested during robustness checks? In other words, are the results robust to alternative classifications of the sand map? If so, please mention this. If not, in the discussion please elaborate on why this is not the case and explain why 80 cm depth is the most relevant depth.

Page 5, line 8: As mentioned earlier, I find it hard to understand and evaluate this section because I do not have a clear overview on the data that were used. Why was it chosen to limit method 1a to Lincolnshire only?

Page 5, line 22: It is not clear to me how the road can be improved by a burst water main. Except due to good reparation of the road.

This section (as well as the section on page 6, line 11) also causes some confusion to me because it reads as if changes in RCI were used to identify bursts of water mains. From later sections I understand that information on burst water mains is available from other sources as well, but I think that this section (page 6) should be rewritten.

Page 7, line 3: What about the directionality of causality here? How can we know that it is the previous burst that causes the next one, and not a common cause, like for example pipes being old, or slow onset landslides? This could be a problem in the current approach.

Page 8, line 1: Like for the two other methodologies, it is important to mention the sources of information here. How many workshops were held, with how many participants? What were their roles and functions and what is the gender balance? Which questionnaire, or form to guide the discussion, has been used? Could this form be added to the appendix?

Page 8, line 6: Did you do these workshops before or after the previous analyses? I.e., did you yourself already have information about the topic during these interviews? How did you make sure not to influence the outcome?

Page 8, line 9: What is meant with a 'thematic analysis'? How was this analysis done? More care needs to be given to the collection and interpretation of the data if social sciences methods are to be used.

Results and discussion

Given the current objective of this manuscript (i.e., the second objective being to test whether a mixed method has an added value), in this section I would not present the results of all methods confounded because that prevents us from appreciating and evaluating the added value of a mixed method approach. I would start by discussing what was learnt from each of the methods, then what would be missed out if the other methods had not been used. Then discuss your overall finding from the integration

of the different methods and use this as an illustration of the value of mixed method approaches.

Page 8, line 16-line 24: Some of these would better fit in the materials and methods section, I think. These lines are still providing information necessary for the interpretation of the results, not yet the results themselves.

Page 9, line 6: It is not clear to me why different scales have been used for the different methodologies. If this one is over whole of England and Wales, why is the other only on one tiny subsection? Please discuss how the difference in scale has a consequence for the comparison of the different methodologies.

Page 12, line 5: Could it be that what is actually measured here is the quality of the reparation works, rather that the impact of the bursts? What is the relevance of this? Wouldn't it make more sense to use municipal data on the costs related to reparation works as a measure of the impact? Secondly, I am not sure about how the spread in road quality after a burst in sandy soils indicates that greater remedial work is required... Also it is not clear whether the difference in spread is significant.

Page 13, line 2: It is not clear to me how you come to this. I think that from the data it is not possible to conclude that the roads are being repaired to a lower standard. the averages did not differ. To my understanding, the following result is also not warranted by the data: "Trenching will also provide preferential hydrological pathways for water compared to the surrounding ground.". How has the analysis led to this result?

Page 14, line 1: It is not clear whether this has been observed in the study, whether it comes from other reports or whether it was mentioned during the workshops. Also the added value to the study is not clear.

Page 14, line 18: I think the findings in this section are worth further elaboration and further study. This could be an interesting added value to literature!

Implications:

Page 18, line 19: To me, it seems that a qualitative assessment was done of the impact, rather than a quantitative assessment.

Minor issues

Title: I would slightly change the title, because its current form is not easy to read. By writing "the influence of . . . on . . . of . . . on . . ." it is not immediately clear the influence of what on what is being studied. After rereading it is clear that you measure the influence of soil on the relation between burst water mains and consequences for society, but this should be clear from the start.

Page 2, line 10: I would appreciate additional explanation for this term which is new to me: "siloed approach".

Page 2, line 21: I am not sure whether this paragraph perfectly fits here. Maybe move it up, just before previous paragraph.

Page 2, line 27: Add apostrophe: organisations'

Page 3, line 13: While I am not an expert in this, I am surprised by this first hypothesis. I would expect a sandy soil to be more porous to water, thus more easily evacuating leaking water and less likely to cause cascading events.

Page 6, line 7: I don't understand what is meant with "To mitigate this spatial inaccuracy, a count of these polygons was used in this context simply to calculate a change in condition."

Figure 4: Please mention total amounts somewhere, not only rates. Also report confidence bars in two subgraphs. In the caption, mention which of the subgraphs is national A, B and unclassified.

Page 13, line 22: (300, 400, 800 homes in media reports) = What is meant by these numbers and could it be possible to add a reference?

Page 16, line 3: "A large water company invested a large amount of money cleaning

sand from the sewers in Lowestoft, only for the sewers to fill up with sand again following the next storm surge." This is an interesting fact, but it doesn't seem relevant for the case study at hand.

Page 19, line 6-9: It is also not totally clear to me how this information helps us to test the hypotheses that were proposed in the introduction.

Page 17, line 17: "This research identified mixed levels of awareness of sand washout risk from infrastructure operators." This is an interesting finding, but it seems to fall outside the scope of the objectives of the manuscript, I think.

Page 18, line 1-2: I wonder, after this study, whether you could say whether it is worth, from a cost-benefit point of view to collect such detailed information?

Page 18, line 31: This was not totally clear to me from the results.

Page 19, line 15: The study did not focus on the UK, but on a much smaller area, I think.

Page 20, line 2: No soil maps were created, to my knowledge.

---

## Author Comment (AC1) · 17 Apr 2018

T. Acland

Thank you for your encouraging comments and helpful suggestions. We will take them on board when revising the manuscript. We will take all your points on board when revising, but due to the work involved, we will not be able to include the cut / clamp analysis, or the revised pipe length estimate in this paper.

In response to your specific points:

1. Noted. We will seek to mention this in our revised manuscript.

[Figure]

2. This will likely be considered, and revisited in a future publication. Thank you. For now we will include your comments in our discussion in this section.

3. We did not analyse the cut/clamp data, but will consider doing so for a future publication.

4. Noted. Thank you.

5. Yes indeed. We have noted this in our personal lives as well, and will mention this in the revised paper.

6. Noted. In the revised manuscript, we will revise our terminology to say that they have a level of engineering reflecting the lower level of traffic use.

Many thanks,

Tim Farewell, Simon Jude, Oliver Pritchard

―――――――――――――――――

---

## Author Comment (AC2) · 17 Apr 2018

K. Mertens

Thank you for the thoughtful and thorough comments and suggestions you have made. We will aim to revise the manuscript taking on board all your comments.

General Comments:

We agree with your comment about testing two hypotheses, so, as you suggest we will revise the manuscript to focus on the core objective of investigating how different soils influence the impact of burst water on society. We will make the objectives / research

questions explicit in the manuscript.

Methods:

We will expand the discussion on the data we have used to provide more detail as requested. Sand map – it was a re-classification of existing maps combining a number of data layers, and we will make this clear in the manuscript. Depth: other depths were investigated in the initial exploratory analysis, but we quickly settled on the 80 cm depth as this is most representative of water pipe depth in the UK. We will remove reference to the other depths in this paper for clarity. Lincolnshire Roads: We only had road data for the one county, so this is why it was limited to this area. Repair: Yes, road surface quality can improve after a burst as it can force remediation of the surface. We will clarify our writing in this area to improve communication. Common cause / cascading failure: Agreed. The data says nothing about cause of the failure, so we are using a mixture of knowledge of system failure mechanisms and data to support our comments in this section. We will clarify this. Workshops: we will describe the workshop design, approach, and analysis in more detail, as requested.

Results and Discussion:

We will consider revising the results approach as you suggest, to consider each method individually, before integrating our discussions. We will move the section as appropriate. Scales of methods: different scales were used due to different data availability. Repairs of Roads / Road Quality: we will clarify our descriptions here. Because we were analysing the road quality in proximal areas which had, and had not been influenced by a burst, the change in RCI, if higher in sandy soils, might reflect that more damage was inflicted on the road, thus requiring more substantial remedial work (rather than a standard cut / replace just to access a burst main.) We will clarify this in our manuscript. We will elaborate / clarify these issues as requested. Implications: Quantitative / qualiatative: yes, that could be true. We used quantitative and qualitative methods, but the integration of these does come to a more qualitative conclusion.
[Figure]

Minor Comments:

Title: we will address this. Thank you. Siloed: Siloes are a grain store – the term means "isolated from each other" or reflects a lack of communication between parties. We will rephrase. Other comments: many of these remaining comments are minor, and we will address them all in the revised manuscript.

Many thanks for your helpful comments. We will revise the manuscript and upload it for review.

Thank you,

Tim Farewell, Simon Jude, Oliver Pritchard

---

## Author Response (AR1)

Response to Reviewers

T. Acland

Thank you for your encouraging comments and helpful suggestions. We have taken them on board when revising the manuscript.

In response to your specific points:

*1. Page 5 Sect 2 Line 10. The sewer data is explained as being of insufficient accuracy. It is also likely that these repairs were detected and resolved much later in time from the initial event. There are lengths of sewer still in operation although in poor condition, and stay that way for a long time.*

Noted. We have mentioned this in our manuscript.

*2. Page 7 Sect 2 line 20. Total mains length will include 'sundries' e.g. hydrant and wash out 'legs', short pipe section associated with valve complexes. The 7% difference in length may be greatly reduced by discounting mains less than something between 1 - 5m in length. You will probably end up spot on!*

As previously discussed, this will likely be considered and revisited in a future publication. Thank you. For now we have included your comments in our discussion in this section.

*3. Page 12 Sect 3 Line 20 and Page 13 Line 6. Are there proportionately more clamp burst repairs in the sand clusters than cut repairs? Cut repairs are usually more serious repairs and are more likely to be responded to quicker as an emergency. Clamp repair bursts may have been running for a much longer time and cause more sub-surface damage.*

We did not analyse the cut/clamp data as many of the bursts data were missing this information, but will consider doing so for a future publication if the data becomes more robust. The alternative, as discussed later in the paper is to consider the impact on individual pipe materials.

*4. Page 14 Sect 3.2.5. This is interesting. The development of wide tolerance repair fittings and the Regulatory/Customer service need for rapid supply restoration may be leading to water companies to repair rather than replace short sections of mains. We have definitions for sewer criticality but no national guidelines for water main criticality. Further there is no nationally agreed sharing of these details between the utility companies.*

Noted. Thank you.

*5. Page 17 Sect 3.3 Line 6. Whilst most hospitals have second/alternative supplies another significant issue is the closure of schools. The impact on students and families will be significant.*

Yes indeed. We have noted this in our personal lives as well, and found examples of this in the media analysis and have mentioned this in the revised paper.

*6. Page 11 Sect 3 Line 5. May be say that minor roads have a more historical or appropriate level of engineering instead of saying they are less well engineered.*

We have revised our terminology to say that they have a level of engineering reflecting the lower level of traffic use.

Many thanks,

Tim Farewell, Simon Jude, Oliver Pritchard

K. Mertens

Thank you for the thoughtful and thorough comments and suggestions you have made. We have revised the manuscript following your general and specific comments.

**General Comments:**

*Main comments General framing: The manuscript is currently presented as a scoping study which has two, very distinct hypotheses which require very different methodologies. On the one hand it is said that the authors want to investigate the relation between soil texture (sand) and the frequency of cross-infrastructure failure. On the other hand they want to test and evaluate a mixed methods approach for this kind of studies. I think these two objectives are too different to fit into one research paper, and certainly so if this research paper does only present the result of a scoping study. The current analysis does not test the second hypothesis. I therefore suggest dropping this second hypothesis and just mention that the purpose is to investigate the influence of soil on the consequences for society by combining different methods. Doing so the manuscript illustrates the value of mixed methods approaches, but does not test a hypothesis as to whether a mixed method is better or worse than another approach. It still allows you to discuss the merits of such an approach in the discussion and conclusion of your research paper. Alternatively, the objective could be to investigate the value of a mixed method approach to study this kind of things. The technical analysis of the correlation with soil can then be presented as one of the different methods or as a case study. It would be good if the objectives and research questions were made explicit. Now, I did only find the hypotheses and had to deduce the objective of the manuscript from it myself.*

We agree with your comment about testing two hypotheses, so, as you suggest we have revised the manuscript to focus on the core objective of investigating how different soils influence the impact of burst water on society.

We have made the objectives / research questions explicit in the manuscript.

*Page 3, line 15: How is this hypothesis tested? By presenting this as a hypothesis it is suggested that the value of this approach will be tested. That is currently not being done.*

We have removed this second hypothesis.

**Methods:**

*Page 3, line 19: I would appreciate a section on the data, or more information on the data at least. How many bursts do you have for your study region (on the map we currently only see those that*

*have led to a cascading event)? How many of these are in sandy soils? What percentage of those that are not on sandy soil have led to cascading events? This comment also holds for the news items. You mention 30+ reports, but is this for 30 different events? How many of these were on sandy soil?... After the section 'methods' I remain with the question whether you have information about all bursts or only about those that led to a cascading event. Do you not have a problem of selection-bias in your sample (eg soil types that have a better water evacuation not leading to cascading events being underreported in the number of bursts)?*

We have included a new section (2) on data. In addition we have reduced discussion of data that was subsequently not used to a negligible level, so there is greater focus on the data that was used. The flow of data has also been clarified in Figure 1.

*Page 4, line 5: I am a little surprised about this sand map. First, at several places in the manuscript it is mentioned that a soil map was created (abstract, or page 20, line 2), while I guess soil maps were already existing. For this manuscript a reclassification of soil groups was just ran based on sand content. I might be wrong, but this does not seem very new to me and it shouldn't be presented as a main contribution of the manuscript. The manuscript makes other contributions. If more than that was done, this should be mentioned here.*

The sand map was a re-classification of existing maps combining a number of data layers, and we have made this clear in the manuscript. We have reduced the emphasis on this part of the work.

*Page 5, line 2: It is mentioned that for consistency sand content at 80 cm depth was used. But were other depths also tested during robustness checks? In other words, are the results robust to alternative classifications of the sand map? If so, please mention this. If not, in the discussion please elaborate on why this is not the case and explain why 80 cm depth is the most relevant depth.*

Other depths were investigated in the initial exploratory analysis, but we quickly settled on the 80 cm depth as this is most representative of water pipe depth in the UK. We have removed reference to the other depths in this paper for clarity.

*Page 5, line 8: As mentioned earlier, I find it hard to understand and evaluate this section because I do not have a clear overview on the data that were used. Why was it chosen to limit method 1a to Lincolnshire only?*

Lincolnshire Roads: We only had road data for the one county, so this is why it was limited to this area. We have also made this more clear by amending Figures 1 and 2. In addition, for clarity we have renamed methods 1a and 1b as Methods 1 and 2. So now Methods 1, 2, and 3 are all geospatial analyses and Method 4 (previously Method 3) is the Workshop / interviews. This also clarifies the flow through the paper in the results section.

*Page 5, line 22: It is not clear to me how the road can be improved by a burst water main. Except due to good reparation of the road. This section (as well as the section on page 6, line 11) also causes some confusion to me because it reads as if changes in RCI were used to identify bursts of water*

*mains. From later sections I understand that information on burst water mains is available from other sources as well, but I think that this section (page 6) should be rewritten.*

Yes, road surface quality can improve after a burst as it can force remediation of the surface. We have re-written this section (p 7 lines 24- page 8 line 2)

*Page 7, line 3: What about the directionality of causality here? How can we know that it is the previous burst that causes the next one, and not a common cause, like for example pipes being old, or slow onset landslides? This could be a problem in the current approach.*

We agree. The data says nothing about cause of the failure, so we are using a mixture of knowledge of system failure mechanisms and data to support our comments in this section. We have mentioned the different possibilities in section 3.2 (page 9, lines 1-4)

*Page 8, line 1: Like for the two other methodologies, it is important to mention the sources of information here. How many workshops were held, with how many participants? What were their roles and functions and what is the gender balance? Which questionnaire, or form to guide the discussion, has been used? Could this form be added to the appendix?*

We have described the workshop design, approach, and analysis in more detail in section 3.4

*Page 8, line 6: Did you do these workshops before or after the previous analyses? I.e., did you yourself already have information about the topic during these interviews? How did you make sure not to influence the outcome?*

These workshops and interviews occurred following initial work in the other three methods. We have clarified the approach to sharing of the results in figure 1, and in section 3.4 (page 10, lines 8-10)

*Page 8, line 9: What is meant with a 'thematic analysis'? How was this analysis done? More care needs to be given to the collection and interpretation of the data if social sciences methods are to be used.*

Our approach has been clarified in Section 3.4, page 10, lies 16-20.

**Results and Discussion:**

*Given the current objective of this manuscript (i.e., the second objective being to test whether a mixed method has an added value), in this section I would not present the results of all methods confounded because that prevents us from appreciating and evaluating the added value of a mixed method approach. I would start by discussing what was learnt from each of the methods, then what would be missed out if the other methods had not been used. Then discuss your overall finding from*

*the integration C4 NHESSD Interactive comment Printer-friendly version Discussion paper of the different methods and use this as an illustration of the value of mixed method approaches.*

We have revised the results section as you suggested, to consider each method individually, before integrating our discussions.

*Page 8, line 16-line 24: Some of these would better fit in the materials and methods section, I think. These lines are still providing information necessary for the interpretation of the results, not yet the results themselves.*

We have moved the section as indicated.

*Page 9, line 6: It is not clear to me why different scales have been used for the different methodologies. If this one is over whole of England and Wales, why is the other only on one tiny subsection? Please discuss how the difference in scale has a consequence for the comparison of the different methodologies.*

We have clarified the difference scales used in the different methods in figures 1 and 2. Different scales were used due to different data availability. As each method is carried out in its own study area the results are not meant to be explicitly compared, but rather, together they build a body of knowledge, from different perspectives around this theme of impacts from burst mains.

*Page 12, line 5: Could it be that what is actually measured here is the quality of the reparation works, rather that the impact of the bursts? What is the relevance of this? Wouldn't it make more sense to use municipal data on the costs related to reparation works as a measure of the impact? Secondly, I am not sure about how the spread in road quality after a burst in sandy soils indicates that greater remedial work is required... Also it is not clear whether the difference in spread is significant.*

We have clarified the scenarios in which we feel the road quality can be change in section 3.1, page 10, lines 23-30). Because we were analysing the road quality in proximal areas which had, and had not been influenced by a burst, the change in RCI, if higher in sandy soils, might reflect that more damage was inflicted on the road, thus requiring more substantial remedial work (rather than a standard cut / replace just to access a burst main.) Because of the very large number of observations, the difference is statistically significant. However, we do recommend caution in drawing strong conclusions from the analysis in isolation (page 11, lines 6-9). This is another benefit of a mixed methods approach.

*Page 13, line 2: It is not clear to me how you come to this. I think that from the data it is not possible to conclude that the roads are being repaired to a lower standard. the averages did not differ. To my understanding, the following result is also not warranted by the data: "Trenching will also provide preferential hydrological pathways for water compared to the surrounding ground.". How has the analysis led to this result?*

This comment was made by a highways engineer in Method 3, and is not derived from the data analysis, so have removed it from this section. This view is supported by comments (p17, lines 13-14)

*Page 14, line 1: It is not clear whether this has been observed in the study, whether it comes from other reports or whether it was mentioned during the workshops. Also the added value to the study is not clear*

This was discussed at the workshop and investigated in subsequent research. The reason we have included this is to highlight that a gas leak is more serious health risk than the original water leak. We have clarified the source of this comment.

*Page 14, line 18: I think the findings in this section are worth further elaboration and further study. This could be an interesting added value to literature!*

Thank you. We will consider a more detailed investigation in the future.

Implications:

*Page 18, line 19: To me, it seems that a qualitative assessment was done of the impact, rather than a quantitative assessment.*

We have rephrased this as requested. We used quantitative and qualitative methods, but the integration of these does come to a more qualitative conclusion.

Minor Comments:

*Title: I would slightly change the title, because its current form is not easy to read. By writing "the influence of . . . on . . . of . . . on . . ." it is not immediately clear the influence of what on what is being studied. After rereading it is clear that you measure the influence of soil on the relation between burst water mains and consequences for society, but this should be clear from the start.*

We have changed the title as requested.

*Page 2, line 10: I would appreciate additional explanation for this term which is new to me: "siloed approach".*

We have removed this unclear term.

*Page 2, line 21: I am not sure whether this paragraph perfectly fits here. Maybe move it up, just before previous paragraph.*

We have moved it as requested.

*Page 2, line 27: Add apostrophe: organisations'*

Done, thank you.

*Page 3, line 13: While I am not an expert in this, I am surprised by this first hypothesis. I would expect a sandy soil to be more porous to water, thus more easily evacuating leaking water and less likely to cause cascading events.*

We have tried to clarify our thinking behind this hypothesis here and throughout the paper.

*Page 6, line 7: I don't understand what is meant with "To mitigate this spatial inaccuracy, a count of these polygons was used in this context simply to calculate a change in condition."*

We have clarified our explanation (p 8, lines 8-11) with reference to the green polygons in Figure 3.

*Figure 4: Please mention total amounts somewhere, not only rates. Also report confidence bars in two subgraphs. In the caption, mention which of the subgraphs is national A, B and unclassified.*

This is now Figure 7. We have clarified what this graph is showing. These are all national assessments from the media analysis.

*Page 13, line 22: (300, 400, 800 homes in media reports) = What is meant by these numbers and could it be possible to add a reference?*

These numbers refer to the number of homes referenced in the media analyses. These references are provided in the Web References section, and Appendix A.

*Page 16, line 3: "A large water company invested a large amount of money cleaning sand from the sewers in Lowestoft, only for the sewers to fill up with sand again following the next storm surge." This is an interesting fact, but it doesn't seem relevant for the case study at hand.*

We have removed this "interesting fact".

*Page 19, line 6-9: It is also not totally clear to me how this information helps us to test the hypotheses that were proposed in the introduction.*

We have removed this.

*Page 17, line 17: "This research identified mixed levels of awareness of sand washout risk from infrastructure operators." This is an interesting finding, but it seems to fall outside the scope of the objectives of the manuscript, I think.*

We have removed this comment

*Page 18, line 1-2: I wonder, after this study, whether you could say whether it is worth, from a cost-benefit point of view to collect such detailed information?*

We have not undertaken a cost benefit analysis here. Given our experience of working with utility companies, changing the collection of any data is a challenge!

*Page 18, line 31: This was not totally clear to me from the results.*

We have tried to clarify our thinking on this point.

*Page 19, line 15: The study did not focus on the UK, but on a much smaller area, I think.*

There was a UK focus for the media analysis and the workshop / interviews, but also with more detailed regional sections. We have adapted the comment to reflect this.

*Page 20, line 2: No soil maps were created, to my knowledge.*

We have reduced the emphasis on this point.

Many thanks for your helpful comments.

Thank you,

Tim Farewell, Simon Jude, Oliver Pritchard

[revised manuscript text omitted]

---

## Author Response (AR2)

Page 3, line 8. It is better to say 'up to' rather than 'less than'.

Thanks for your comment. I actually, feel that "less than" is more appropriate than "up to" in this case, as we are talking about the coverage of soils over England and Wales, so I have left it as it is.

Page 11, para 4.2. This leads into a significant conclusion and I feel that this para may be made more robust to support the conclusion. The result for sandy soils burst repeats is new knowledge and may be more prominent. Further these details are open to criticism from any less informed readers. By using repeats in up to the 10m/10d criterion there is the risk of including reworked jobs following premature failure of the first repair. The data used has been reviewed by several parts of that business including contract payments as work carries a guarantee the data is checked to ensure no repeat work is included; all data used were independent repairs. This fact may need to be emphasised and hence the result and significant conclusion are fully valid.

Thank you. I have included the following sentence to make this point:

*The bursts data used shows only independent repairs, so jobs to repair previous repairs which have failed prematurely are excluded.*

Page 17, lines 11 & 12. I don't understand this sentence. At least a word is missing.

I have rephrased it as follows:

*Multiple media reports described how small road surface deformations were initially misdiagnosed and treated as simple surface failures, only for a larger deformation or hole to appear the next day.*

Page 20, line 21. Whilst historically true, there is a lot of new technology being applied to infrastructure monitoring right now that is expected to be cost effective. This sentence may be rephrased.

Yes, there are development in this area, yet even pressure is not monitored across whole networks. I have rephrased this as follows:

*Monitoring of infrastructure stability can incur substantial costs and is often unfeasible across an entire network, so reactive responses to infrastructure failures are common.*

Page 21, line 15. The data for the water industry is already available in the National Failures Database (NFD) held by the UK Water Industry Research organisation (UKWIR). Further development of the analytical methodology proposed by this paper requires similar data from Highways and other infrastructure industries.

I have included the following sentences to address these points.

*UK wide data on water mains bursts is being collected in the National Failures Database, held by the UK Water Industry Research Organisation. Similar databases for other infrastructure communities would be of value.*

Page 21, line 27. Societal impacts in some ways are recorded within the water industry. The duration of supply interruptions is another performance measure (e.g. the old DG3) that is reportable to OFWAT, so data is available. In fact it is now two measures, one is properties off water for a long time and the other is proportion of properties off water by minutes.

Thank you, I have included the following sentence:

[revised manuscript text omitted]